# COSUTI: A Core Outcome Set (COS) for Interventions for the Treatment of Uncomplicated Urinary Tract Infection (UTI) in Adults

**DOI:** 10.3390/antibiotics11121846

**Published:** 2022-12-19

**Authors:** Claire Beecher, Sinead Duane, Akke Vellinga, Andrew W. Murphy, Martin Cormican, Andrew Smyth, Patricia Healy, Michael Moore, Paul Little, Carmel Geoghegan, Declan Devane

**Affiliations:** 1HRB Trials Methodology Research Network, School of Nursing and Midwifery, University of Galway, H91 TK33 Galway, Ireland; 2School of Nursing and Midwifery, University of Galway, H91 TK33 Galway, Ireland; 3Discipline of Marketing, J.E. Cairnes School of Business and Economics, University of Galway, H91 TK33 Galway, Ireland; 4School of Public Health, Physiotherapy and Sports Science, University College Dublin, D04 V1W8 Dublin, Ireland; 5Discipline of General Practice, HRB Primary Care Clinical Trial Network Ireland, College of Medicine, Nursing and Health Sciences, University of Galway, H91 TK33 Galway, Ireland; 6Discipline of General Practice, School of Medicine, University of Galway, H91 TK33 Galway, Ireland; 7Discipline of Bacteriology, School of Medicine, University of Galway, H91 TK33 Galway, Ireland; 8HRB Clinical Research Facility Galway, University of Galway, H91 TK33 Galway, Ireland; 9Primary Care Population Sciences and Medical Education, Faculty of Medicine, University of Southampton, Southampton SO17 1BJ, UK; 10PPI Ignite, University of Galway, H91 TK33 Galway, Ireland; 11Evidence Synthesis Ireland and Cochrane Ireland, School of Nursing and Midwifery, University of Galway, H91 TK33 Galway, Ireland

**Keywords:** core outcome set, urinary tract infections, cystitis, Delphi survey, consensus, PPI

## Abstract

*Background:* Uncomplicated urinary tract infections (UTIs) are among the most common presentations of bacterial infections in the outpatient setting. The variation of outcomes reported in trials to assess the most effective treatment interventions for uncomplicated UTIs has meant that comparing and synthesising the outcomes across trials is challenging and limits the reliability of evidence which would otherwise inform healthcare decisions. *Objective:* Develop a Core Outcome Set (COS) for interventions for the treatment of uncomplicated UTIs in otherwise healthy adults. *Methods:* The COS development consisted of three phases: (1) A systematic review to identify outcomes reported in randomised trials and systematic reviews of randomised trials comparing the effectiveness of any interventions for the treatment of uncomplicated UTI in otherwise healthy adults; (2) Outcomes identified in the systematic review were prioritised in an online 3-round modified Delphi survey with healthcare practitioners (*n* = 68), researchers (*n* = 5), and people who have experienced or cared for someone experiencing a UTI (*n* = 180); (3) An online consensus meeting to determine the final COS with healthcare practitioners and policymakers (*n* = 9), researchers (*n* = 4), and people who have experienced or cared for someone experiencing a UTI (*n* = 7). *Results:* We identified a large number of outcomes. Through the use of robust consensus methods, those outcomes were reduced to a core set of six outcomes that should, at a minimum, be measured and reported in randomised trials and systematic reviews of interventions treating uncomplicated UTIs in adults.

## 1. Introduction

Urinary tract infections (UTIs) are among the most common bacterial infections in the outpatient setting [1]. At most ages, UTIs are much more common in women than men [2]. It has been estimated that by the age of 24, a third of the female population will have experienced at least one UTI that received antimicrobial treatment [3]. Clinically, UTIs may be categorised as complicated or uncomplicated.

A UTI is considered complicated when it is associated with a structural or functional abnormality that compromises the urinary tract [4]. In contrast, an uncomplicated UTI typically affects an otherwise healthy individual without known structural or functional abnormalities of the urinary tract [5].

Empirical antimicrobial therapy is often used to treat uncomplicated UTIs [6] and, as such, uncomplicated UTIs are a significant cause of antimicrobial prescription worldwide, with wide variation in the range and dosage of antimicrobials used internationally [7]. This treatment has, over time, contributed to the rise in antimicrobial resistance in uropathogens such as Escherichia coli, the organism most associated with UTI [1,8].

The continued focus on antimicrobial stewardship [9] has driven an increase in the volume of clinical trials investigating various interventions for the treatment of UTIs (which can vary from antibiotics to drinking more fluids). However, the variation of outcomes being reported in these clinical trials makes it difficult to make direct comparisons between the effectiveness of a particular treatment intervention between trials. This results in difficulties in the conduct of evidence synthesis and meta-analysis which limit the availability of strong evidence which would inform healthcare decisions [10].

The development of a core outcome set (COS) addresses this issue. A COS is the minimum set of outcomes that should be measured and reported in all clinical trials in specific areas of health or health care [11]. It is also suitable to use a COS in clinical audits or research other than randomised trials, such as observational studies [12]. Researchers may wish to include other outcomes relevant to their research, but the COS should always be included as a minimum [13]. Once a COS has been developed, additional work is then required to achieve consensus on how the COS should be measured [14].

To date, a COS for interventions for the treatment of uncomplicated UTIs in adults has not been developed.

The objective of the COSUTI study was to develop a COS for interventions to treat uncomplicated UTIs in adults to address the issues identified above.

The COSUTI research team was led from Ireland and the study was conducted between February 2017 and September 2021.

## 2. Materials and Methods

The COSUTI study was registered prospectively with the COMET initiative (registration number 950 and available online at https://www.comet-initiative.org/Studies/Details/950, accessed on 15 March 2022). The protocol for the study has been published, which includes detailed methods [15]. The findings are reported here in line with the Core Outcome Set-STAndards for Reporting statement (COS STAR) [16].

COSUTI consisted of three phases: 1. a systematic review was conducted to identify outcomes comparing the effectiveness of interventions for treating uncomplicated UTI in adults; 2. outcomes were presented to stakeholders for prioritisation in an online Delphi survey, which was repeated three times; 3. an online consensus meeting was held with stakeholders to determine which outcomes would be included in the final COS.

### 2.1. Phase 1: Systematic Review

A systematic search was conducted to identify a comprehensive list of outcomes reported in randomised trials and systematic reviews of randomised trials between 2007–2017 comparing the effectiveness of interventions for the treatments for uncomplicated UTI in adults. The Cochrane Database of Systematic Reviews, PubMed, and Embase were searched. All outcomes identified were extracted verbatim, and similar outcomes were grouped into domains and subdomains. The methods and results of the systematic review have been published in full [17].

### 2.2. Phase 2: Delphi Survey

A three-round modified Delphi survey was conducted to achieve consensus on the importance of the outcomes identified in the systematic review. The outcomes from the systematic review were assessed by the steering group to combine outcomes and minimise overlap between domains. Many outcomes varied in the time points at which the outcomes were measured and reported and the steering group standardised these time points where appropriate.

Participants were purposively sampled from stakeholder groups most impacted by interventions for the treatment of uncomplicated UTIs in adults:People who have experienced or cared for someone experiencing a UTIGeneral practitionersHospital-based medical practitionersNurses or midwivesResearchersPharmacistsMicrobiologistsPolicymakers

Round 1 participants were recruited with emails to publicly available accounts and professional organisations, through personal networks of the steering group, and a social media campaign. Participants were asked to share the survey invitation with others who they thought had the relevant experience to participate. Demographic questions were included in the survey to monitor the geographical spread of participants and stakeholder groups, which was then used to focus additional recruitment where needed. A wide geographical spread of participants was sought to ensure that the resulting COS would be acceptable to, and used by, research groups from countries of varying income levels.

No additional participants were recruited in rounds 2 and 3.

The survey was conducted using web-based Delphi software by Calibrum [18]. Each round ran for four weeks. Following the consent and demographics sections, participants were asked to rate the level of importance of each outcome on a Likert scale from 1 (for an outcome that is not important) to 9 (for an outcome that is critically important). A plain English explanation was presented for each outcome. Participants were advised that when an outcome included the term ‘treatment’, this referred to a management decision ranging from antibiotics to increased fluids. Participants were advised to click the ‘unable to score’ option if they felt unable to comment on the level of importance of any of the outcomes.

Following scoring of the outcomes in round 1 of the Delphi survey, participants were asked if there were any important outcomes that they felt were missing and should be considered. Any outcome suggested by more than one participant was brought forward to round 2. An open question was also included at the end of the round 1 asking participants if they had anything else to add.

Responses to the open question indicated that the volume of outcomes included was considered too much. In response, a post hoc change was made to reduce the volume of outcomes presented in round 2. An outcome was brought to round 2 of the Delphi survey if >60% of all respondents rated it 7 or greater, or if an average rating of 7 or more was received from people who have experienced or cared for someone experiencing a UTI. An additional four outcomes suggested by round 1 participants were also included.

All round 1 participants were sent an invitation to participate in round 2, and a reminder was sent each week to complete. Participants were asked to rescore each outcome and the score allocated by them in round 1 was highlighted. Participants were also presented with three graphs illustrating the scores given by each stakeholder group for each outcome to consider (see Figure 1). For the purpose of presenting the findings of the Delphi survey to participants in rounds 2 and 3, the stakeholder groups involved were combined into three groups, i.e., healthcare providers, researchers, and people or carers with experience of UTI.

Outcomes were included in round 3 if they were rated as critical (rated 7–9) by at least 70% of respondents. To ensure that public-prioritised outcomes were not overwhelmed by other groups, any outcome with an average score of 7 or more by people who have experienced or cared for someone experiencing a UTI was also included in round 3.

All round 2 participants were sent an invitation to participate in round 3 and a reminder was sent each week to complete. A similar graphic display as for round 2 was shown for each outcome.

Participants in round 3 were asked to submit their email if they were interested in participating in the consensus meeting.

### 2.3. Phase 3: Consensus Meeting

The consensus meeting to finalise the COS took place online using the Zoom [19] software programme on the 7 September 2021. The decision to hold the consensus meeting online was determined by the COVID-19 pandemic.

Outcomes rated as critical (rated 7–9) by at least 70% of participants and of limited importance (rated 1–3) by fewer than 15% of participants in round 3 of the Delphi survey were considered to meet the definition of ‘consensus in’ and were brought to the consensus meeting. Outcomes that were rated of limited importance (rated 1–3) by at least 70% of participants and rated as critical (rated 7–9) by less than 15% of participants in round 3 were considered ‘consensus out’ and were not discussed at the meeting. Any outcomes that did not meet either definition were classified as ‘no consensus’ and were discussed at the consensus meeting.

Prior to the meeting, each participant received an information pack to inform the discussion that would take place. These packs included an agenda, a detailed participant guide, a list of the outcomes for discussion, plain English definitions, and the round 3 Delphi survey results categorised by stakeholder group and the accompanying graphs as per Figure 1.

Based on prior experience of steering group members facilitating online consensus meetings, it was decided that a maximum of 20 participants would be optimal. Recruitment for the meeting was focused on securing a broad geographical and stakeholder spread. A wide geographical spread of participants was again sought to ensure that the resulting COS would be relevant and acceptable to stakeholders from countries of varying income levels. In addition to Delphi-survey participants who expressed their interest in attending, authors of prominent research papers on the topic and individuals with relevant experience suggested by the steering group were contacted and invited to attend. Three steering group members were asked to attend the meeting as participants and as a source of information, if needed, on the processes adopted throughout the study.

Polls were used to decide on inclusion or exclusion of outcomes, and outcomes were included in the COS if agreed on by 70% or more of consensus-meeting participants.

## 3. Results

### 3.1. Phase 1: Systematic Review

The systematic review included 41 papers and identified 124 outcomes that could be categorised across 18 domains, highlighting the heterogeneity of outcome reporting in this area.

### 3.2. Phase 2: Delphi Survey

Following the review and refinement of the results of the systematic review by the steering group, 116 outcomes were deemed eligible for inclusion in round 1 of the survey (Appendix A). A total of 253 participants from 26 countries took part in round 1, of whom 27% (*n* = 68) were healthcare practitioners, 2% (*n* = 5) were researchers, and 71% (*n* = 180) were people who had experienced or cared for someone experiencing a UTI.

Most participants were from England (*n* = 135), with representation also secured from countries including Scotland (*n* = 14), Ireland (*n* = 35), Ukraine (*n* = 5), Uganda (*n* = 1), USA (*n* = 2), Norway (*n* = 2), Australia *n* = 9), New Zealand (*n* = 2), and Brazil (*n* = 1).

In round 2, participants were presented with the 60 outcomes that remained following the application of the round 1 exclusion criteria. Four new outcomes that round 1 participants identified were also presented in round 2, i.e., the use of alternative treatment to antibiotics, patient-reported impact of a UTI on the relationship with a partner, cost of UTI treatment, and impact on mental health wellbeing.

Round 2 was completed by 76% (*n* = 192) of those who had completed round 1. Of the 192 participants, 26% (*n* = 50) were healthcare practitioners, 4% (*n* = 7) were researchers, and 70% (*n* = 135) were people who had experienced or cared for someone experiencing a UTI.

In round 3 of the survey, participants were presented with the 43 outcomes that remained following the application of the round 2 exclusion criteria. Round 3 was completed by 64% (*n* = 123) of those who had completed round 2. Of the 123 participants, 27% (*n* = 33) were healthcare practitioners, 4% (*n* = 5) were researchers, and 69% (*n* = 85) were people who had experienced or cared for someone experiencing a UTI.

### 3.3. Phase 3: Consensus Meeting

All 43 outcomes included in round 3 of the Delphi study were eligible for discussion at the consensus meeting as all were classified as ‘no consensus’ following analysis.

Twenty participants from 11 different countries attended the consensus meeting, i.e., Cameroon, Canada, England, Indonesia, Ireland, Kenya, Nigeria, Norway, Pakistan, Scotland, and Uganda. A total of 45% (*n* = 9) of those who attended were healthcare practitioners or policymakers, 20% (*n* = 4) were researchers, and 35% (*n* = 7) were people who had experienced or cared for someone experiencing a UTI.

At the start of the meeting, the participants acknowledged the large volume of outcomes for discussion. Although each outcome in itself was judged through the Delphi as important to measure, the focus of the meeting was to decide upon how essential it is for each outcome discussed to be measured and reported in all clinical trials for interventions for the treatment of uncomplicated UTI in adults.

Following discussion of each individual outcome, and through voting by participants using polls, six outcomes were deemed eligible for inclusion in the final COS (Table 1). Participants agreed on the definitions for each of the outcomes with minor editing post-meeting for wording clarity.

The need to combine several similar outcomes prioritised in the Delphi survey into a single outcome that could feasibly be measured was evident at the meeting. One example of this was in relation to the outcome ‘Recurrence of UTI symptoms following initial resolution in the first 28 days after the start of treatment’. This outcome is the result of the decision by participants to combine outcomes that had been prioritised in the Delphi survey within the ‘Relapse’ domain. Table 2 presents the outcomes that were combined to create this outcome.

## 4. Discussion

The COSUTI study developed a COS for interventions to treat uncomplicated UTIs in adults. The COS includes six core outcomes that are of most importance to people who have experienced or cared for someone experiencing a UTI, General Practitioners, hospital-based medical practitioners, nurses, midwives, researchers, pharmacists, microbiologists, and policymakers.

Uncomplicated UTIs are among the most common bacterial infections in the outpatient setting [1] and are a significant cause of antimicrobial prescription internationally, with wide variation in the range and dosage of antimicrobials used [7]. The European Association of Urology clearly highlights within its 2022 guidelines the critical importance that must continue to be placed on antimicrobial stewardship [9].

The ever-growing focus on antimicrobial stewardship is attributable to the increased volume of clinical trials being undertaken to investigate the various interventions for the treatment of uncomplicated UTIs. The conduct of the COSUTI study is timely and the COS developed will make it easier for findings of the research being conducted in this area to be compared and combined [12].

### 4.1. Interpretation

The systematic review conducted as phase one of this project [17] found that many of the outcomes identified varied in the time points at which they were to be measured and reported. Variance in timepoints is common throughout the COS-development literature, as demonstrated by Young et al. who found that the timing of outcome assessment was reported in one-third of the 132 COS-development studies that they identified, and of those, more than half reported outcomes with different time points as unique outcomes [20]. Methodological guidance regarding the management of this heterogeneity when developing a COS, however, is lacking.

The use of outcome time points in the COS was the focus of much conversation at the consensus meeting, particularly in outcomes 1–3, and, ultimately, the need for a time point was considered by participants on an individual basis for each of these outcomes. For example, throughout the discussions that focused on combining the outcomes presented in Table 2, the need for an ‘anchor’ or time point within this outcome dominated. Justifications ranged from the possibility of including no time point, which could facilitate the pooling of a wide range of results from clinical trials, to the inclusion of a time point which would both make it clear how long after treatment patients needed to be followed and discourage the conduct of short-term trials and premature reporting of results. The latter justification was, on balance, considered most important by participants. Conversely, in relation to outcome 3, ‘Worsening or progression of UTI symptoms’, the value of an ‘anchor’ for the outcome was raised, but discussions ultimately did not support it.

### 4.2. Strengths and Limitations

A major strength of the COSUTI project was the participation of people who have experienced a UTI or cared for someone who has experienced a UTI. Representation from this stakeholder group in particular was critical and a key element of the recruitment drive for both the Delphi survey and the consensus meeting. We are confident that the high volume of participants from this group increases the validity of the COS and has led to the development of a patient-centric COS reflective of the significant input of patients and the public.

For a COS to achieve the goal of facilitating the synthesis of findings from different studies, it must be acceptable to and used by research groups in different countries [13] The online format of the Delphi survey and a targeted recruitment campaign facilitated the involvement of participants from 26 countries. Similarly, the online consensus meeting allowed participation from 11 different countries, adding to the generalisability of results.

The inclusion of participants from countries of different income levels also increased the applicability of the COS internationally, which was acknowledged during the discussions at the consensus meeting. For some outcomes, participants from lower- to middle-income countries raised the issue that, due to resources and guidelines in existence in their country, the measurement of the outcome would simply not be possible, even though the outcome could be of importance.

Another benefit of including a geographical spread of participants in the meeting was in relation to the wording of the COS, in particular outcome 3, which had originally included the term ‘complicated UTI’. This term was interpreted differently in the participating countries and, therefore, reworded to ensure universal understanding.

One of the limitations of this Delphi study was the length of the survey round 1. The systematic review identified 124 outcomes. Although these outcomes were standardised by time point and combined where possible, 116 outcomes were included in round 1. We received feedback from some participants in the ‘anything else to add’ question following the first round that the survey was too long. Considering this feedback, a post hoc change was made to reduce the volume of outcomes presented in round 2 by 49%, which resulted in the presentation of 60 outcomes in round 2.

Another limitation was that we were not able to include a representative from microbiologists or policymakers in the survey. However, they were represented at the consensus meeting.

## 5. Conclusions

The volume of clinical trials investigating various interventions for the treatment of uncomplicated UTIs in adults continues to increase. However, a COS for this specific health area had not been developed prior to the COSUTI study.

Through robust consensus methods and engagement with participants from key stakeholder groups internationally, a COS comprising six outcomes has been developed for interventions for treating uncomplicated UTIs in adults.

We encourage funders, researchers, and the scientific community to adopt this COS as the minimum set of outcomes that should be measured and reported in all clinical trials for interventions for the treatment of uncomplicated UTIs in adults. They are also suitable for use in clinical audits or research other than randomised trials.

The uptake of the COS presented will play a key role in aiding future evidence synthesis and meta-analysis to inform health care decisions.

## Figures and Tables

**Figure 1 antibiotics-11-01846-f001:**
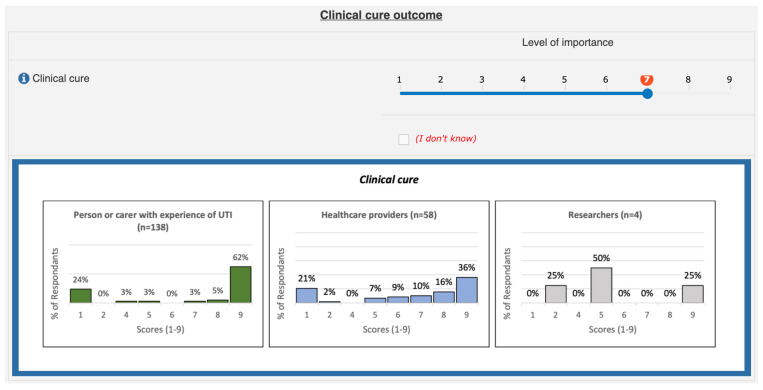
Presentation of round 1 Delphi survey results.

**Table 1 antibiotics-11-01846-t001:** Final COS to be included in future research on interventions for treating uncomplicated UTI in adults.

No.	Outcome	Definition
1	Time (days) from initiation of treatment to resolution of symptoms	The number of days it takes, from the time a treatment * is started until a person is no longer experiencing symptoms of an uncomplicated UTI. Symptoms are experienced or felt by the person themselves and indicate that they might have a UTI, for example, a strong, persistent urge to pass urine or a burning sensation when passing urine
2	Recurrence of UTI symptoms following initial resolution in the first 28 days after start of treatment	UTI symptoms return within 28 days from the time treatment * was started, after they had initially been resolved
3	Worsening or progression of UTI symptoms	The symptoms of a person’s initial episode of a UTI worsen as distinct from a recurrence of a UTI where UTI symptoms return after they had initially been resolved
4	Person’s self-reported quality of life	A person’s quality of life, as reported by the person themselves
5	Person’s satisfaction with the treatment of the UTI	A person’s level of contentment or happiness with the treatment * of their UTI
6	Adverse events/effects	Outcomes occurring as an unintended consequence of the treatment (e.g., vomiting, abdominal pain, nephrotoxicity)

* A ‘treatment’ is a management decision ranging from, for example, antibiotics to drinking more fluids.

**Table 2 antibiotics-11-01846-t002:** Outcomes combined to create outcome 2.

Original Outcomes	Final Outcome
Frequency of relapses until day 28 after initial resolution of symptoms	Recurrence of UTI symptoms following initial resolution in the first 28 days after start of treatment
Relapse after initial resolution of symptoms	
Early relapse of symptoms by day 14 after initial resolution of symptoms	
Frequency of relapse or new infection by day 15 after initial resolution of symptoms	
Frequency of relapse or new infection by day 28 after initial resolution of symptoms	
Incidence of new symptoms of UTI after initial clinical cure	
Recurrent UTI after initial resolution of symptoms up to day 28	
Recurrence 4–6 weeks following treatment completion	
Clinical recurrence of signs after initial resolution of symptoms	
Recurrence of a UTI within 6 months after initial resolution of symptoms	
Clinical recurrence within 30 days	
Clinical recurrence within 90 days	

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
