# Peer review of "COSUTI: A Core Outcome Set (COS) for Interventions for the Treatment of Uncomplicated Urinary Tract Infection (UTI) in Adults"

_antibiotics, 2022, doi:10.3390/antibiotics11121846_

Round 1

Reviewer 1 Report

Very well presented and executed study. 

Author Response

Thank you for your review of the COSUTI paper- we appreciate the feedback you have provided.  

Reviewer 2 Report

Beesher et al., have attempted a very nice study and it could be very helpful in combating AMR too. The manuscript fits well in the scope of journal and can be processed further for publication after revision. The manuscript needs some changes before proceeding further. My Comments are:

1.     First of all, the authors have mentioned the adult patients, in my opinion here they should mention age group as they outcomes of UTIs in immunocompromised patients, old age and young age could be different.

2.     The material and methods part is confusing and needs an extensive revision.

3.     Mention study area/setting and duration.

4.     Line 30-31: Abstract: The objective need to revise. May be “The current study was design to develop a Core Outcome Set (COS) for interventions for the treatment of uncomplicated UTIs in adult patients”.

5.     Line 32-41: Rather then to write in point, write as one paragraph. Rephrase the sentence at line 32 as “The development of COS was consisted of three phases”

6.     Line 42-45: What kind (domain) of outcome authors want to describe here? Authors need to define healthcare practitioners and policymakers, researchers in the material and methods section of manuscript.

7.     Line 78-80: More this paragraph in discussion section and replace it with detailed objective and need of study.

8.     Line 95-100: Why authors used the already published systematic review from 2007-2017 to be enrolled in current study? Why not until 2021 or 2022? As during the pandemic there was significant differences were observed in the treatment of other kind of infections.

9.     Furthermore, authors need to mention here more about the protocol of systematic review.

10.  Line 96, 107, 109: I didn’t understand the word like “3ategorize”?

11.  Section 2.2 and 2.3 are unclear.

12.  What was the criteria to choose the target population and how they make sure the biasness as it was an online survey.

13.  What was the tool of survey, what type of questions they asked.

14.  What was the criteria if target population has refused to take part in the survey?

15.  Line 214-216: Here again, mention the outcomes (at least domains), which outcome you want to mention here?

16.  Line 222: these people were hospital staff or patient attendants?

17.  Line 281: Need to correct the reference “project17”

18.  The discussion part needs some more data.

19.  In the discussion part, authors can discuss their results with current strategies and guidelines.

20.  In the discussion section, there is only 1 reference. The authors can cite more relevant guidelines and strategic studies and can discuss their results with them.

21.  The strengths and limitation’s part is too long, need to concise.

Reviewer 3 Report

Kindly improve the sentencing, using punctuation and spelling in a few places.

Please explain the basis for the selection of countries for the study.

 Explain the measures taken to check the reliability of the online data collection for the study

Strengthen the concluding message and stress the importance of your study.

Round 2

Reviewer 3 Report

Thanks for considering the corrections. Hope the publication has the potential to publish.